# Clinical and Demographic Factors Associated with Receiving an Opioid Prescription following Admission to the Pediatric Intensive Care Unit

**DOI:** 10.3390/children9121909

**Published:** 2022-12-06

**Authors:** Amy L. Holley, Trevor A. Hall, Ben Orwoll, Anna C. Wilson, Eleanor A. J. Battison, Denae Clohessy, Cydni N. Williams

**Affiliations:** 1Division of Pediatric Psychology, Department of Pediatrics, Oregon Health & Science University, Portland, OR 97239, USA; 2Pediatric Critical Care and Neurotrauma Recovery Program, Oregon Health & Science University, Portland, OR 97239, USA; 3Division of Pediatric Critical Care, Department of Pediatrics, Oregon Health & Science University, Portland, OR 97239, USA; 4Department of Medical Informatics and Clinical Epidemiology, Oregon Health & Science University, Portland, OR 97239, USA

**Keywords:** pediatric intensive care, pain, opioid

## Abstract

Opioids are commonly used to treat pain in the pediatric intensive care unit (PICU), and many children receive opioid prescription(s) at discharge. The frequency of opioid prescriptions at discharge and associations with individual characteristics and clinical factors are unknown. This study aimed to identify (1) the number of children who receive an opioid prescription at PICU discharge and (2) the demographic and clinical factors associated with receiving an opioid prescription. Data were collected via the electronic medical record. The sample was 3345 children (birth to 18 years) admitted to the PICU and discharged to home or an inpatient rehabilitation setting. In total, 23.7% of children were prescribed an opioid at discharge. There were group differences in who received opioid prescriptions (yes/no) related to PICU diagnosis, length of hospital stay, number of days on mechanical ventilation, number of previous hospitalizations, organ dysfunction score, and admission type (surgical versus non-surgical). Binary logistic regression models examined predictors of opioid prescription at discharge for the total sample and diagnostic subgroups. Older age and surgical admission type were the most consistent predictors of receiving an opioid prescription. Future research should examine prescription usage patterns and how use of opioids is associated with pain and functional outcomes over time.

## 1. Introduction

Among critically ill children in the pediatric intensive care unit (PICU), opioids are used to treat acute pain as a component of anesthesia for procedures, and for comfort during mechanical ventilation. Recent guidelines from the Society of Critical Care Medicine highlight the need for the critical care community to minimize dose and exposure to opioids in children, given risk for important morbidities in and after the PICU. For some children, prolonged use of opioids in the PICU is, nonetheless, necessary as part of acute treatment; however, prolonged opioid use is associated with withdrawal syndromes, sometimes necessitating slow post-discharge tapers of long-acting medications, such as methadone [1]. Opioids are also frequently prescribed at discharge for moderate to severe pain associated with children’s injury or illness [2]. Previously reported data from a PICU follow-up clinic showed one in five children reported clinically significant pain one to three months after hospital discharge [3]. Rates of opioid use in PICU patients after discharge are unknown, but, given the injury and illness severity associated with PICU hospitalization, pain and withdrawal symptoms post-admission are likely common, and place these children at risk for negative consequences of opioid use.

While opioids are a necessary part of treatment for some children, it is also important to highlight the potential risks associated with opioid use in the outpatient setting. Critical to pediatric populations is the fact that the effects of opioids on the developing brain are poorly understood. Importantly, receipt of opioids during PICU admission is a significant risk factor for worse cognitive outcomes after PICU discharge [4]. Specifically, opioids are associated with worse outcomes in comparison to healthy controls or normed population data across cognitive domains of general intelligence, attention, processing speed, executive functioning, memory, visual motor integration, and motor development. For children exposed to opioids both during and post-PICU admission, impact on the developing brain can be even more significant from prolonged use.

Of additional importance is the impact of opioid use over time. Opioids are not ideal for long-term pain control, and additional risks of prolonged use in pediatric populations (e.g., sedation and respiratory depression, misuse and abuse) have been identified, compounding risks of long-term morbidity associated with PICU admission [1,5]. Opioid availability in the home after discharge due to excess opioid prescribing and/or unsafe storage can also be associated with adverse effects [5]; the potential for misuse can be exacerbated in those with concurrent mental health disorders [6]. Research on post-intensive care syndrome in pediatrics (PICS-p) has highlighted the high rates of anxiety, depression, and post-traumatic stress in children after PICU admission, and these mental health concerns may put this population at additional risks for opioid misuse [7].

In addition, unused prescription opioids in pediatric post-hospitalization and post-surgery samples are common, with recent studies indicating many families have leftover medication after they stop giving the child the prescription [8,9,10]. Risks of opioids remaining in the home can include accidental overdoses, intentional ingestions, or diversion to other family members or the community. Understanding which children receive prescriptions can inform recommendations and guidelines for follow-up and targeted intervention among the PICU population to reduce prolonged opioid exposure and risk of misuse in this vulnerable population.

Data on the frequency of opioid prescriptions at PICU discharge are limited and, given the milieu of factors potentially contributing to opioid-related complications, understanding how prescribing relates to individual characteristics (age, sex at birth) and clinical factors (e.g., diagnosis, length of hospital stay, use of mechanical ventilation) is important. To fill these gaps, this study aimed to (1) identify the proportion of children who receive an opioid prescription at time of hospital discharge after PICU hospitalization using data from a single academic medical center, and (2) describe the demographic and clinical factors associated with receiving an opioid prescription.

## 2. Materials and Methods

This study was conducted at an academic medical center in the northwestern United States. The PICU has 20 beds, approximately 1500 admissions per year, and provides quaternary care services including Level 1 Pediatric Trauma. Study procedures including a waiver of consent were approved by the Institutional Review Board. Data were collected via the electronic medical record (EMR) from children who had been admitted to the PICU between 1 January 2018 and 30 June 2021 (referred to as the study period).

Inclusion criteria were: (a) age at PICU admission between birth and 18 years, (b) the first PICU admission documented in electronic medical record during the study period, (c) patient was alive at time of discharge, (d) discharge location was home or inpatient rehabilitation setting.

### 2.1. Measures

All data were extracted from the electronic medical record (EMR). Accuracy of key variables (opioid prescriptions, discharge disposition, admission diagnoses) were reviewed by study staff to ensure reliability and categorize for analysis.

#### 2.1.1. Demographics

Data collected included child’s date of birth (used to calculate age at time of PICU admission), sex at birth, race, and ethnicity. Data on race and ethnicity were abstracted from the medical record, based on routine collection methods and according to Oregon’s REALD (Race, Ethnicity, Language and Disability) scheme [11]. Children were classified as Hispanic if Hispanic ethnicity was selected. Children with other ethnic identities were categorized as non-Hispanic. If the family declined to provide race or ethnicity or there were no data available, participants were categorized as “unknown”. Given small sample sizes in race categories, the sample was categorized as white/Caucasian versus other races to permit comparison in statistical analyses.

#### 2.1.2. Primary PICU Diagnosis

The primary diagnosis for each PICU admission was assigned prospectively by the treating intensivist and, after individual chart review, was coded as one of ten primary diagnosis categories: cardiac, endocrine, fluids/electrolytes/nutrition/gastrointestinal (FEN-GI), hematology/oncology, ingestion, neurologic, other surgical (e.g., orthopedic, plastic, transplant surgeries), respiratory, sepsis/infection, or trauma (including traumatic brain injury, accidental, and non-accidental trauma).

#### 2.1.3. Opioid Prescription at Discharge

Youth were coded (yes/no) regarding whether or not they were given an opioid prescription at discharge as the primary study outcome. Opioids were defined as Drug Enforcement Agency (DEA) Schedule II medications classified as opioid analgesics.

#### 2.1.4. Organ Dysfunction

Pediatric Logistic Organ Dysfunction-2 (PELOD-2) is a quantitative scoring system for organ dysfunction in children based on assessments of five organ systems. The PELOD-2 has been validated in critically ill children and closely correlates with mortality [12]. In addition to a total PELOD score, data indicate mortality is related to PELOD score in a non-linear fashion, with scores of 10 and higher associated with greater mortality. PELOD scores were calculated based on each child’s worst values for each organ system during the entire hospitalization and reported both in terms of a total score and categorically with PELOD scores grouped as follows: 0–1, 2–9, and ≥10 [13].

#### 2.1.5. Length of Hospital Stay

Calculated in days from hospital admission to hospital discharge. This includes days in the PICU and days on the medical floors.

#### 2.1.6. Mechanical Ventilation

Calculated as the number of calendar days during which a patient received invasive mechanical ventilation at any point during the day.

#### 2.1.7. Primary Surgical Admission

Youth were coded (yes/no) according to whether or not their PICU admission was primarily due to having a surgical procedure requiring post-operative monitoring in PICU.

#### 2.1.8. Previous PICU Admissions and Hospitalizations

The count of previous times children in this study had previously been in the PICU or been hospitalized as listed within the institution’s medical record system prior to the onset of the study. (Note: EMR data are only available starting 2009, so admissions prior to 2009 could not be counted).

#### 2.1.9. Functional Status at PICU Admission

The Functional Status Scale (FSS) assesses five domains of functioning (mental status, sensory, communication, motor function, and feeding), with each domain scored on a 5-point scale ranging from 1 (normal) to 5 (very severe dysfunction) [14]. The FSS for this study was calculated by a PICU attending physician at admission to reflect baseline function prior to acute injury or illness associated with current hospitalization [15]. FSS scores range from 6 to 30, with higher scores indicating poorer function. FSS total scores of 6–7 are considered “normal” in the current sample with 8 and higher categorized as “impaired”. Note: the FSS score was available for *n* = 3217 children as it was not listed in the EMR for *n* = 128 participants.

### 2.2. Statistical Analysis

Data were analyzed using SPSS v.28 (IBM, Armonk, NY, USA). Summary statistics were used to describe characteristics of the sample and are reported in Table 1. Means and standard deviations were used for continuous data, and categorical items were described using frequency statistics. Chi-square/*t*-tests were used to examine group differences (opioid prescription at discharge versus no opioid prescription at discharge) by demographic (e.g., age, sex at birth) and clinical factors (e.g., primary PICU diagnosis, length of hospital stay, days on a ventilator, PELOD score). A multivariable logistic regression examined predictors of opioid prescribing at discharge (yes/no).

## 3. Results

### 3.1. Sample Descriptives

There were 3345 children (ages birth–18 years) admitted to the PICU during the study period who met inclusion criteria. Participant demographics and sample descriptives are located in Table 1. Mean age was 6.98 years (SD = 6.17) and sex at birth was 55.1% male. Average length of hospital stay was about one week (M = 8.12 days, SD = 19.40). In total, 24.6% children were on a ventilator during their admission (M = 1.62 days, SD = 9.72). For 29.0% of children, admission was related primarily to a surgical procedure.

Average FSS at admission was 6.93 (SD = 2.54, range 6–28) indicating grossly normal baseline function (FSS = 6–7) for 84.0% of the sample. Average maximum PELOD score was 4.00 (SD = 4.07, range 0–29). Percentages of children falling into PELOD categories were: score 0–1, 19.9%; score 2–9, 69.1%; score 10 or greater, 11.0%. For 87.1% of the sample, this was their first PICU hospitalization, and for 69.8%, the PICU admission was their first hospitalization.

Most frequent primary diagnosis categories associated with PICU admissions were: respiratory (23.9%), neurologic (16.6%), cardiac (15.2%), endocrine (9.8%), and trauma (9.3%). The majority of children were discharged home (98.0%), with 2.0% discharged to a rehabilitation facility.

### 3.2. Opioids at Discharge: Associations with Clinical and Demographic Factors

In total, 23.7% of children received an opioid prescription at discharge. Chi-square tests were used to reveal differences in opioid prescribing by primary diagnosis (*χ^2^*(9, *n* = 3345) = 740.71, *p* < 0.001) and that children were more likely to be discharged with an opioid prescription if the PICU stay was associated with a primary surgical admission (*χ^2^*(1, *n* = 3345) = 1212.18, *p* < 0.001). For descriptive purposes, percentages of children who received an opioid prescription at discharge within each primary diagnostic group, by age group (0–1, 2–4, 5–10, 11–15, and 16–18 years), and if the admission was for a primary surgery are located in Table 2. Youth with cardiac (30%), neurologic (27.2%) and hematology-oncology (13%) had the highest rates of being discharged on opioids. No children admitted for an ingestion received an opioid prescription at discharge. Notably, 8.9% of children admitted for an ingestion had an opioid-related ingestion.

Analyses of chi-square tests were also used to examine associations between receipt of opioid at discharge and clinical and demographic factors. PELOD score category was also associated with opioid prescription at discharge (*χ^2^*(2, *n* = 3345) = 97.48, *p* < 0.001), with children in higher PELOD score categories having greater proportions of children discharged on opioids. Similarly, children with “impaired” versus “normal” FSS scores at baseline were more likely to receive an opioid prescription at discharge (*χ*^2^(1, *n* = 3215) = 5.85, *p* = 0.016). There were no differences in receipt of an opioid prescription by discharge location (i.e., home or rehab facility), sex at birth, ethnicity (Hispanic versus Non-Hispanic) or race (categorized as white/Caucasian versus other races).

Results of *t*-tests examining group differences by opioid prescription at discharge (yes/no) and continuous variables are located in Table 3. Results revealed children who received an opioid prescription had longer hospital admissions, greater number of days on mechanical ventilation, and higher numbers of previous hospitalizations than those not discharged on opioids. There were no differences in opioid prescribing by number of previous PICU admissions.

Given that this study took place over a 4-year period which spanned the COVID-19 pandemic, analyses also examined associations between year of PICU admission (2018, 2019, 2020, 2021) and both receipt of an opioid prescription (yes/no) and number of days on mechanical ventilation using chi-square and ANOVA, respectively. Results revealed no significant differences in percentages of children prescribed opioids by study year nor associations between study year and number of days on a ventilator.

Binary logistic regression models were used to examine predictors of opioid prescription at discharge for the total sample and by primary PICU diagnosis.

A binary logistic regression using the total sample was performed to ascertain the effects of patient age, sex at birth, functional status (“normal” versus “impaired”), organ dysfunction score category, days on mechanical ventilation, number of previous hospitalizations, and if the PICU stay was related to a primary surgical admission on the likelihood that children would be prescribed opioids at discharge (Table 4). The logistic regression model was statistically significant (*χ^2^*(8, *n* = 3345) = 1163.35, *p* < 0.001). The model explained 45.4% (Naglekerke R^2^) of the variance in opioid prescribing and correctly classified 84.2% of the cases. In the total sample, children with a primary surgical admission were significantly more likely to receive opioids than children hospitalized for primary medical diagnoses. Additional significant predictors of receiving opioids at discharger were age (older), functional status (impaired at baseline), organ dysfunction score (higher), and number of days on mechanical ventilation (greater) (Table 4).

The same regression model covariates used for the total sample were used to run models within four of the diagnostic categories that had the largest numbers of children prescribed opioids (cardiac, hematology-oncology, neurologic, trauma) (Table 5). The sole significant predictor of receiving an opioid prescription at discharge for all four of the diagnostic groups was if the PICU admission was related to a primary surgical procedure. Age predicted opioid prescription for cardiac, hematology-oncology, and trauma groups; in the neurologic sample associations between age and opioid prescription approached significance (*p* = 0.06). In the neurologic sample, baseline impaired functional status predicted opioid prescribing. In the trauma sample, sex (female) and PELOD score category (greater organ dysfunction) were also significant predictors for receiving an opioid prescription at discharge. Notably, number of days on a ventilator and number of previous hospitalizations were not significantly associated with opioid prescribing at discharge among any of the four subgroups.

## 4. Discussion

The results of this study contribute to the limited data on rates of opioid prescribing at PICU discharge and clinical and demographic factors associated with receiving an opioid prescription. The data obtained include both percentages of who received an opioid prescription by diagnostic group but also how individual (e.g., age, sex) and PICU stay-related factors (e.g., days on mechanical ventilation, organ dysfunctions score) were associated with opioid prescribing. Specifically, findings revealed that 23.7% of children received an opioid prescription at PICU discharge.

The strongest predictor of being discharged on an opioid post-PICU admission was being admitted for a primary surgical intervention, with these children (both in the total sample and individual diagnostic groups) more likely to receive an opioid prescription at discharge than those with non-surgical-related admissions. Our study highlights that children admitted to the PICU for surgical procedures should potentially receive closer follow-up with continued opioid education and medication management post-discharge, which is a key component of care plans. Given the risks associated with opioid use and misuse, guidance on how to safely discontinue these medications in a timely fashion is paramount.

Another important finding from this descriptive study is data showing how opioid prescribing varied by age within each primary diagnostic category (Table 2). Results showed age emerged as an important factor associated with opioid prescribing with results of the logistic regression showing older age was associated with increased likelihood of being prescribed an opioid at discharge. That said, among some diagnosis groups (e.g., cardiac, respiratory, neurologic), infants and young children (0–1 year) had the highest rates of opioid prescriptions and were more likely to receive opioid prescriptions than older children within that diagnostic category. The high rates of opioid prescribing in young children in these samples is not surprising, given that many infants admitted to the PICU with these diagnoses are born with cardiac defects or conditions that require surgical intervention in the first year of life. Moreover, in this age group, opioids often need to be prescribed at discharge for weaning children who were on prolonged mechanical ventilation during their PICU stay. The effects of opioids on the developing brain are poorly understood, but data do show opioids to be directly neurotoxic and they are associated with worse cognitive outcomes [4]. The risk of opioid use across ages with differential effects on neurodevelopment and outcome depending on age of exposure is a burgeoning field of study, and further assessing trajectories of opioid use post-PICU discharge in this sample is warranted [4].

The large number of children who receive opioids at PICU discharge also highlights the need for additional education to children, parents, and providers regarding opioid use outside the hospital setting. Going home on an opioid is an important component of pain management following a hospitalization for a surgery, serious illness, or injury. That said, opioid availability can also be associated with risks. Many children prescribed opioids do not use the whole prescription and leftover medications are not regularly disposed of or kept in a secure environment (e.g., lockboxes) [16,17]. This poses risks for intentional and accidental ingestions for the patient and their household members in the future. Given that data from the current study were collected from the EMR we do not have data on usage (e.g., number of pills taken, leftover medications, other household factors that could impact use and misuse). Future studies should assess post-discharge factors that could impact risk to patients and families (e.g., parental monitoring, medication storage, understanding of prescription instructions), with the goal of informing interventions targeting children discharged from the PICU with an opioid prescription and their parents.

Similarly, given results highlighting subsamples of children discharged from the PICU have an increased likelihood of being prescribed opioids, efforts to improve coordination of care post-PICU discharge with children’s primary care provider and outpatient specialty team taking over the child’s medication management could be increased. This could include additional conversations between PICU providers and teams taking over the child’s medical care regarding rationale for the opioid prescription and suggested plans for medication discontinuation. If a surgical subspecialty is involved during the admission, discussions between the PICU team and subspecialty providers regarding opioid prescribing are likely ongoing, so developing a collaborative plan for opioid management and discontinuation may be feasible. Additionally, continued development of local- and national-level guidelines related to opioid prescribing and management in PICU survivors should be considered. Research suggests that opioid prescribing practices and consumption following common pediatric surgical procedures vary by region of the country [8,9,10], so additional efforts to standardize opioid prescribing may be warranted.

Strengths of this study include the large sample size and use of a broad sample of children with a spectrum of medical diagnoses akin to the typical case mix of PICU admissions. Moreover, using the EMR can increase data accuracy (e.g., not relying on participant reports regarding receipt of an opioid prescription) and garner additional details related to the child’s PICU care (e.g., FSS and PELOD scores, number of days on mechanical ventilation) not typically available via self- or parent-report.

Importantly, there are some weaknesses of this study. First, data on opioids at discharge were coded only as yes/no. Our analysis did not evaluate the potential role of opiate type or dosing, or assess other non-pharmacologic pain management strategies that could have impacted opioid use in the hospital or providers’ decision to prescribe opioids at discharge. Given the wide variability in opioids and other pain management interventions children in the PICU receive, this will be an important future direction to examine. This analysis can also examine how individual clinical and demographic factors are not only associated with whether or not children receive an opioid prescription, but how they relate to type, dosing, and usage patterns.

An additional limitation is data on pain, psychological factors or other indicators of function (e.g., disability, quality of life) that could not be examined, given use of solely EMR data. It is possible that self-report of pain during PICU admission and at time of discharge influences whether children are prescribed opioids. This is likely a larger factor in children who are able to rate and self-report pain (e.g., school age and older), and how ability to request pharmacological relief impacts opioid prescribing should be examined in future research.

This study is also limited by measures of premorbid function. While we were able to include history (total number) of PICU admissions and hospitalizations, the baseline physician-assigned FSS, and the PELOD score in regression models, greater details on children’s medical history (e.g., history of illness, injury or surgeries, chronic pain diagnoses) relevant to health status could not be examined. Moreover, we do not have information on opioid use prior to admission. We recognize it is possible that patients’ historical use of opioids can impact prescribing patterns and participant usage. It is also possible that other factors that can impact prescribing (e.g., opioid prescription at or prior to admission, patient medication allergies, history of substance use disorders, mood symptoms) played a role in whether or not children were discharged with a prescription. This can be further examined in prospective studies that include patient and family report data.

Moreover, while we do have some data on race and ethnicity of participants, we are limited by how information was categorized for the medical record (e.g., both race and ethnicity categories included “Hispanic”) and if children/families chose to provide this information at time of admission. Given the large amount of missing data and the way data were categorized, we could not include these variables in regression models. An important future direction will be to assess if opioid prescribing patterns following PICU discharge are different for children of different races and ethnicities and, if so, how these prescribing differences are associated with other individual- (e.g., diagnosis) and provider-level factors.

A final important consideration is related the sample. Trauma admissions aged ≥15 years are most often treated in the adult ICU at our institution, so percentages of children admitted to the PICU for trauma may be lower than other samples. Moreover, the EMR data analyzed were collected partly during the COVID-19 pandemic and patient-level characteristics (e.g., primary diagnoses) are likely different than pre-pandemic due to restrictions on elective surgeries and changing disease epidemiology during the pandemic. Despite these limitations and considerations, this study provides an important contribution to the literature and what we know about characteristics of children who receive opioids at discharge post-PICU admission.

## 5. Conclusions

Nearly one in four PICU survivors in our multiyear sample received an opioid prescription at discharge to treat pain or facilitate weans from prolonged usage. Youth undergoing surgical procedures are at particularly high risk for opioid prescribing at discharge and represent a population for targeted intervention. Other risk factors varied by admission diagnosis category. Given risks associated with opioid use and misuse, providers should prioritize medication discontinuation and pursue alternative treatment strategies for pain, as well as increase education for parents and providers with these goals in mind.

## Figures and Tables

**Table 1 children-09-01909-t001:** Demographic and clinical characteristics of the sample *n* = 3345.

Child Demographics	
**Age, M (SD)**	6.99 (6.17)
**Sex assigned at birth, *n* (%)** Male Female	1842 (55.1%) 1503 (44.9%)
**Child ethnicity, *n* (%)** Hispanic Non-Hispanic Unknown/Not reported	622 (18.6%) 2333 (69.7%) 390 (11.7%)
**Child race, *n* (%)** American Indian/Alaska Native Asian Black Hispanic/Latino/a/x Middle Eastern/North African Native Hawaiian/Pacific Islander White Multiple Races Unknown/Not reported	50 (1.5%) 86 (2.6%) 78 (2.3%) 43 (1.3%) 3 (0.1%) 47 (1.4%) 2344 (70.1%) 324 (9.7%) 370 (11.1%)
**PICU Diagnosis Category, *n* (%)** Cardiac Endocrine FEN-GI Hematology-Oncology Ingestion Neurologic Respiratory Sepsis Infection Trauma Other Surgical	509 (15.2%) 329 (9.8%) 147 (4.4%) 191 (5.7%) 293 (8.7%) 555 (16.6%) 801 (23.9%) 125 (3.7%) 310 (9.3%) 85 (2.5%)

SD, standard deviation;
PICU, pediatric intensive care unit; FEN-GI, fluids/electrolytes/nutrition/gastrointestinal.

**Table 2 children-09-01909-t002:** Descriptive data of children who received an opioid prescription by age and primary diagnosis.

	Cardiac *N* = 240	Neurologic *N* = 216	Hem/Onc *N* = 105	Trauma *N* = 83	Respiratory *N* = 61	Other Surgical *N* = 56	Sepsis/ Infection *N* = 17	FEN–GI *N* = 15	All *N* = 793
0–1 years	96	108	6	9	26	9	1	4	*n* = 259
Surgical Medical	91 5	106 2	3 3	1 0	16 10	9 0	0 1	3 1	
2–4 years	66	17	11	12	10	6	3	3	*n* = 128
Surgical Medical	66 0	17 0	7 4	3 9	8 2	6 0	1 2	3 0	
5–10 years	36	33	32	21	10	8	3	3	*n* = 146
Surgical Medical	33 3	26 7	23 9	5 16	3 7	8 0	1 2	3 0	
11–15 years	26	34	37	39	9	29	7	3	*n* = 184
Surgical Medical	21 5	26 8	34 3	6 33	4 5	29 0	0 7	1 2	
16–18 years	16	24	19	2	6	4	3	2	*n* = 76
Surgical Medical	14 2	21 3	12 7	0 2	3 3	4 0	0 3	2 0	

Total sample = 3345, total recipients of opioids = 794 (23.7%); “Endocrine” had *n* = 1 and was removed from table.

**Table 3 children-09-01909-t003:** Group differences on clinical factors by opioid prescription (yes/no) at discharge.

	Opioids—Yes		Opioids—No		*t* (3348)	*p*	Cohen’s *d*
	M	SD	M	SD			
Length of hospital stay (days)	9.75	26.59	7.62	16.51	−2.71	0.007	0.110
Days on mechanical ventilation	2.43	13.91	1.37	7.96	−2.68	0.007	0.109
Previous hospitalizations	5.33	12.76	3.50	11.56	−3.82	<0.001	0.155
Previous PICU admissions	0.34	1.18	0.34	1.36	−0.125	0.92	0.005

**Table 4 children-09-01909-t004:** Binary logistic regression model for total sample estimating effects of clinical and demographic factors on receipt of opioid at PICU discharge.

	OR	Lower 95%	Upper 95%	*p*
Age	1.05	1.03	1.07	<0.001
Sex (female)	0.87	0.71	1.07	0.19
Functional status (impaired)	0.75	0.56	0.996	0.046
Organ dysfunction				
Score 2–9	1.79	1.29	2.48	<0.001
Score ≥ 10	1.41	0.92	2.17	0.118
# Days on a ventilator	1.01	1	1.02	0.024
# Previous hospitalizations	0.99	0.97	1	0.157
Primary surgical admission	24.97	19.94	31.27	<0.001

# number; OR—Odds Ratio.

**Table 5 children-09-01909-t005:** Binary logistic regression models estimating effects of clinical and demographic factors on receipt of opioid at PICU discharge by individual diagnostic group (cardiac, neurologic, trauma, hematology-oncology).

	Cardiac (*n* = 509)				Neurologic (*n* = 555)				Trauma (*n* = 310)				Hematology-Oncology (*n* = 191)			
	OR	*p*	Lower 95%	Upper 95%	OR	*p*	Lower 95%	Upper 95%	OR	*p*	Lower 95%	Upper 95%	OR	*p*	Lower 95%	Upper 95%
Age	1.09	0.002	1.03	1.15	1.05	0.06	1.00	1.10	1.16	<0.001	1.10	1.23	1.11	0.001	1.04	1.18
Sex (female)	0.75	0.18	0.50	1.14	1.12	0.69	0.64	1.96	2.29	0.01	1.25	4.18	0.76	0.47	0.37	1.59
FSS (impaired)	0.90	0.71	0.52	1.56	0.45	0.03	0.22	0.91	2.63	0.57	0.09	75.38	0.67	0.54	0.18	2.45
PELOD (2–9)	0.90	0.85	0.30	1.68	1.77	0.19	0.75	4.17	2.90	0.01	1.30	6.45	0.34	0.12	0.09	1.33
PELOD (≥10)	0.54	0.28	0.17	1.67	0.29	0.20	0.04	1.94	7.43	0.01	1.84	29.87	0.49	0.51	0.06	4.01
# Days on Ventilator	1.01	0.25	0.99	1.03	1.00	0.97	0.85	1.19	0.97	0.75	0.82	1.15	0.97	0.72	0.84	1.12
# Previous Hospitaliz.	1.01	0.39	0.99	1.03	0.98	0.10	0.96	1.00	0.73	0.09	0.51	1.05	0.99	0.28	0.98	1.01
Primary surgical admit	23.72	<0.001	11.98	47.00	65.11	<0.001	35.78	118.49	9.52	<0.001	3.23	28.11	8.72	<.001	4.14	18.36

# number; FSS, functional status score; PELOD, Pediatric Logistic Organ Dysfunction.

## Data Availability

The data presented in this study are available on request from first author Amy Holley. The data are not publicly available due to detailed information about the participants contained in the data.

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
