# Peer review of "Clinical and Demographic Factors Associated with Receiving an Opioid Prescription following Admission to the Pediatric Intensive Care Unit"

_children, 2022, doi:10.3390/children9121909_

Round 1

Reviewer 1 Report

Thank you for submitting this interesting paper exploring relationships between the outcome of receiving an opioid prescription at discharge from the pediatric intensive care unit and various clinical and demographic characteristics. Your team aimed to answer an important clinical question: Among children admitted to the pediatric intensive care unit at a large medical center between January 1, 2018 and June 30, 2021, what clinical and demographic factors predict prescription of an opioid medication at discharge?   

Please consider the following recommendations to strengthen and clarify your manuscript: 

Introduction: 

1. The introduction section begins by outlining relationships chronic pain and opioid use, yet the variable of chronic pain diagnosis is not evaluated in this manuscript. While lack of acute pain measures are explicitly stated as a limitation to the study in the discussion section, there is no mention regarding why past history of chronic pain diagnosis was not included as an important factor. 

2. The study inclusion criteria spans over a year of the COVID-19 pandemic, where increased numbers of ventilated patients in the critical care units were observed. The introduction section does not mention this as a potential factor influencing opioid prescription patterns, however. Is there rationale for not mentioning this important time in history with regard to ICU patients? 

Setting and participants: 

1. Provide rationale for why the sample included individuals from birth to 18 years of age, given that ability to articulate pain and request pharmacological relief is very different across the age spectrum. 

2. Provide rationale for why only the first hospital encounter of the selected timeframe was chosen for inclusion in the analysis? Was there a significant proportion of repeat hospitalizations during this time? 

3. Provide rationale for examining only half of the year 2021 in this manuscript? 

Data collection: 

  1. Please clarify whether data were abstracted primarily for the purpose of this project, or whether an existing database of abstracted data was queried. If abstracted primarily for this project, describe why abstraction versus extraction techniques were used. 

Data analysis: 

  1. 1. Given the impacts of COVID and national changes to evidence-based practice guidelines regarding opioid prescribing, it may be useful to provide an overview of the proportion of patients discharged each year (2018, 2019, 2020, 2021) with an opioid to describe whether year of hospitalization impacted opioid prescriptions  

  1. 2. This statement appears to contain a typo: Youth with ingestion (0.0%) related admissions had the lowest likelihood of receiving an opioid prescription at discharge. 

  1. 3. Table 3 lists Cohen’s d values that are extremely large (typically these range from 0.1 - 2.0?). Can you clarify whether these values are accurate?  

  1. 4. Clarify why length of stay is not included in Table 4 as a covariate, given significant bivariate differences noted in Table 3.  

  1. 5. Table 5 is referenced but not shown. In the text describing this table, it sounds like four, separate regression models were conducted to validate findings from the overall sample. Is this correct? If so, please provide your rationale for this approach versus including these diagnoses as predictor variables in the overall model as reported in Table 4 with dichotomous flags (i.e. cardiac_flag defined as 1 or 0= yes/no). 

Limitations: 

1. Presence of an opioid medication prior to hospital admission may be a significant predictor of continued prescription at discharge. Please list the lack of this variable as a limitation. 

2. Many limitations are listed, but few strengths of the study are mentioned. Please provide some strengths of the study design here. 

Conclusion: 

  1. In this section, a clinical implication is offered to increased provider and patient education to pursue alternative treatment strategies for pain, yet the project did not include multi-modal pharmacological pain management strategies as a possible predictor for discharge prescription with opioid. Please either include this variable as a predictor to support the statement (i.e. create flags for whether other non-opioid pain medications were prescribed at discharge and add into your analytical models) or report this construct in the introduction and discussion sections as an important consideration that may reduce need for opioids at discharge. The lack of inclusion of this variable could then be mentioned in the limitation section.

Author Response

Reviewer #1

Comment 1: The introduction section begins by outlining relationships chronic pain and opioid use, yet the variable of chronic pain diagnosis is not evaluated in this manuscript. While lack of acute pain measures are explicitly stated as a limitation to the study in the discussion section, there is no mention regarding why past history of chronic pain diagnosis was not included as an important factor. 

                Response 1:  Past history of chronic pain was not included as a variable in analysis as this is not standardly reported in the EMR. We have added a statement to the discussion indicating potential importance of past chronic pain history in opioid use/prescribing and add this as an important factor to consider in future longitudinal studies (see page 14).

Comment 2: The study inclusion criteria spans over a year of the COVID-19 pandemic, where increased numbers of ventilated patients in the critical care units were observed. The introduction section does not mention this as a potential factor influencing opioid prescription patterns, however. Is there rationale for not mentioning this important time in history with regard to ICU patients? 

Response 2:  While it is recognized numbers of ventilated patients on adult critical care units, this was not consistently observed in pediatric intensive care units so no a prioi hypothesis was made about how COVID19 may impact opioid prescribing. That said, at the reviewer’s recommendation we now include data in the results section that compares # of days on a ventilator and percentage of youth were prescribed opioids (yes/no) by study year (2018, 2019, 2020, 2021). As shown, neither number of days on a ventilator nor % of youth prescribed opioids are significantly different across study years (see pages 9-10).

Comment 3:  Provide rationale for why the sample included individuals from birth to 18 years of age, given that ability to articulate pain and request pharmacological relief is very different across the age spectrum. 

                Response 3:  Given dearth of data on opioid prescribing at PICU discharge we elected to utilize a full pediatric sample. We further highlight in the discussion that age and ability to articulate pain and request medications during admission and at discharge may be associated with prescribing patterns and this can be further explored in future prospective longitudinal studies (see page 14).

Comment 4. Provide rationale for why only the first hospital encounter of the selected timeframe was chosen for inclusion in the analysis? Was there a significant proportion of repeat hospitalizations during this time? 

                Response 4:  The first PICU admission was chosen because we were interested in patient outcomes following their first experience in the PICU. A portion of youth had subsequent PICU admissions during the time period extracted (e.g., 13.7% two PICU admissions; 4.66% three PICU admissions). Examining patterns of opioid prescribing over time in the context of multiple PICU admissions is outside the focus of the current study but can be examined in future research.

Comment 5:  Provide rationale for examining only half of the year 2021 in this manuscript? 

Response 5: Only ½ of 2021 was included as that is when data was pulled from the EMR for data cleaning and analysis.   

Comment 6: Please clarify whether data were abstracted primarily for the purpose of this project, or whether an existing database of abstracted data was queried. If abstracted primarily for this project, describe why abstraction versus extraction techniques were used. 

Response 6:  Data were extracted from the medical record specifically for this project. Chart level review was used to confirm diagnoses extracted from the medical record in the data pull. We have clarified this in the methods section (see pages 5-6).

Comment 7: Given the impacts of COVID and national changes to evidence-based practice guidelines regarding opioid prescribing, it may be useful to provide an overview of the proportion of patients discharged each year (2018, 2019, 2020, 2021) with an opioid to describe whether year of hospitalization impacted opioid prescriptions  

Response 7:  See response to Comment 2. As stated in that response, at the reviewer’s recommendation we ran a chi-square analysis examining opioid prescribing at discharge (yes/no) by admission year. We did not find any significant differences. This is now included in the manuscript (see pages 9-10).

Comment 8:  This statement appears to contain a typo: Youth with ingestion (0.0%) related admissions had the lowest likelihood of receiving an opioid prescription at discharge. 

Response 8:  Thank you for the close read and attention to detail. This number is actually accurate. It reflects that zero youth who were admitted for an ingestion were prescribed an opioid at discharge. This is not unexpected given the reason for admission.

Comment 9:  Table 3 lists Cohen’s d values that are extremely large (typically these range from 0.1 - 2.0?). Can you clarify whether these values are accurate?  

Response 9:  Thank you for catching this error. The correct Cohen’s d values are now presented in Table 3.  

Comment 10: Clarify why length of stay is not included in Table 4 as a covariate, given significant bivariate differences noted in Table 3.  

Response 10:  Length of stay was not included as a covariate given collinearity with other factors included in the model (e.g.,  PELOD score, # of days on a ventilator) which are known to be related to the severity of injury or illness. These factors were selected instead of # of days on a ventilator as we know that hospital-related factors (e.g., bed availability, provider comfort level with patient disposition) and social factors (e.g., transportation, availability of medical equipment in home) can impact length of stay.

Comment 11: Table 5 is referenced but not shown. In the text describing this table, it sounds like four, separate regression models were conducted to validate findings from the overall sample. Is this correct? If so, please provide your rationale for this approach versus including these diagnoses as predictor variables in the overall model as reported in Table 4 with dichotomous flags (i.e. cardiac_flag defined as 1 or 0= yes/no). 

Response 11:  We apologize that Table 5 was not visible with the documents you reviewed. It was uploaded with the submission and we are hopeful the reviewer can see it with the revision. We elected to conduct the analysis using this method because we felt it was inappropriate to use particular diagnostic group as a reference group (e.g., results would be interpreted as “compared to youth with X diagnoses, youth with Y diagnosis were more likely to be prescribed opioids). In addition, running models by diagnostic group allowed us to examine if predictors were differently associated with opioid prescribing by diagnostic group (as shown in results presented on page 10 and in Table 5).  

Comment 12: Presence of an opioid medication prior to hospital admission may be a significant predictor of continued prescription at discharge. Please list the lack of this variable as a limitation. 

Response 12:  The reviewer makes an important point and we now include presence of an opioid medication prior to hospital admission as a limitation (see page 14).

Comment 13: Many limitations are listed, but few strengths of the study are mentioned. Please provide some strengths of the study design here. 

Response 13:  Thank you for this comment. Strengths have been added. These include large sample size, use of a broad sample akin to the typical case mix of PICU admissions, and extraction of data from the EMR (see page 13).

Comment 14: In this section, a clinical implication is offered to increased provider and patient education to pursue alternative treatment strategies for pain, yet the project did not include multi-modal pharmacological pain management strategies as a possible predictor for discharge prescription with opioid. Please either include this variable as a predictor to support the statement (i.e. create flags for whether other non-opioid pain medications were prescribed at discharge and add into your analytical models) or report this construct in the introduction and discussion sections as an important consideration that may reduce need for opioids at discharge. The lack of inclusion of this variable could then be mentioned in the limitation section.

Response 14:  The reviewer brings up an important point – no we do not have data on other multi-modal pharmacological pain management strategies that participants could have been using during their PICU admission may have impacted opioid use or prescription at discharge. We have added this as a limitation to the discussion section (see page 13).

Reviewer #1

Comment 1: The introduction section begins by outlining relationships chronic pain and opioid use, yet the variable of chronic pain diagnosis is not evaluated in this manuscript. While lack of acute pain measures are explicitly stated as a limitation to the study in the discussion section, there is no mention regarding why past history of chronic pain diagnosis was not included as an important factor. 

                Response 1:  Past history of chronic pain was not included as a variable in analysis as this is not standardly reported in the EMR. We have added a statement to the discussion indicating potential importance of past chronic pain history in opioid use/prescribing and add this as an important factor to consider in future longitudinal studies (see page 14).

Comment 2: The study inclusion criteria spans over a year of the COVID-19 pandemic, where increased numbers of ventilated patients in the critical care units were observed. The introduction section does not mention this as a potential factor influencing opioid prescription patterns, however. Is there rationale for not mentioning this important time in history with regard to ICU patients? 

Response 2:  While it is recognized numbers of ventilated patients on adult critical care units, this was not consistently observed in pediatric intensive care units so no a prioi hypothesis was made about how COVID19 may impact opioid prescribing. That said, at the reviewer’s recommendation we now include data in the results section that compares # of days on a ventilator and percentage of youth were prescribed opioids (yes/no) by study year (2018, 2019, 2020, 2021). As shown, neither number of days on a ventilator nor % of youth prescribed opioids are significantly different across study years (see pages 9-10).

Comment 3:  Provide rationale for why the sample included individuals from birth to 18 years of age, given that ability to articulate pain and request pharmacological relief is very different across the age spectrum. 

                Response 3:  Given dearth of data on opioid prescribing at PICU discharge we elected to utilize a full pediatric sample. We further highlight in the discussion that age and ability to articulate pain and request medications during admission and at discharge may be associated with prescribing patterns and this can be further explored in future prospective longitudinal studies (see page 14).

Comment 4. Provide rationale for why only the first hospital encounter of the selected timeframe was chosen for inclusion in the analysis? Was there a significant proportion of repeat hospitalizations during this time? 

                Response 4:  The first PICU admission was chosen because we were interested in patient outcomes following their first experience in the PICU. A portion of youth had subsequent PICU admissions during the time period extracted (e.g., 13.7% two PICU admissions; 4.66% three PICU admissions). Examining patterns of opioid prescribing over time in the context of multiple PICU admissions is outside the focus of the current study but can be examined in future research.

Comment 5:  Provide rationale for examining only half of the year 2021 in this manuscript? 

Response 5: Only ½ of 2021 was included as that is when data was pulled from the EMR for data cleaning and analysis.   

Comment 6: Please clarify whether data were abstracted primarily for the purpose of this project, or whether an existing database of abstracted data was queried. If abstracted primarily for this project, describe why abstraction versus extraction techniques were used. 

Response 6:  Data were extracted from the medical record specifically for this project. Chart level review was used to confirm diagnoses extracted from the medical record in the data pull. We have clarified this in the methods section (see pages 5-6).

Comment 7: Given the impacts of COVID and national changes to evidence-based practice guidelines regarding opioid prescribing, it may be useful to provide an overview of the proportion of patients discharged each year (2018, 2019, 2020, 2021) with an opioid to describe whether year of hospitalization impacted opioid prescriptions  

Response 7:  See response to Comment 2. As stated in that response, at the reviewer’s recommendation we ran a chi-square analysis examining opioid prescribing at discharge (yes/no) by admission year. We did not find any significant differences. This is now included in the manuscript (see pages 9-10).

Comment 8:  This statement appears to contain a typo: Youth with ingestion (0.0%) related admissions had the lowest likelihood of receiving an opioid prescription at discharge. 

Response 8:  Thank you for the close read and attention to detail. This number is actually accurate. It reflects that zero youth who were admitted for an ingestion were prescribed an opioid at discharge. This is not unexpected given the reason for admission.

Comment 9:  Table 3 lists Cohen’s d values that are extremely large (typically these range from 0.1 - 2.0?). Can you clarify whether these values are accurate?  

Response 9:  Thank you for catching this error. The correct Cohen’s d values are now presented in Table 3.  

Comment 10: Clarify why length of stay is not included in Table 4 as a covariate, given significant bivariate differences noted in Table 3.  

Response 10:  Length of stay was not included as a covariate given collinearity with other factors included in the model (e.g.,  PELOD score, # of days on a ventilator) which are known to be related to the severity of injury or illness. These factors were selected instead of # of days on a ventilator as we know that hospital-related factors (e.g., bed availability, provider comfort level with patient disposition) and social factors (e.g., transportation, availability of medical equipment in home) can impact length of stay.

Comment 11: Table 5 is referenced but not shown. In the text describing this table, it sounds like four, separate regression models were conducted to validate findings from the overall sample. Is this correct? If so, please provide your rationale for this approach versus including these diagnoses as predictor variables in the overall model as reported in Table 4 with dichotomous flags (i.e. cardiac_flag defined as 1 or 0= yes/no). 

Response 11:  We apologize that Table 5 was not visible with the documents you reviewed. It was uploaded with the submission and we are hopeful the reviewer can see it with the revision. We elected to conduct the analysis using this method because we felt it was inappropriate to use particular diagnostic group as a reference group (e.g., results would be interpreted as “compared to youth with X diagnoses, youth with Y diagnosis were more likely to be prescribed opioids). In addition, running models by diagnostic group allowed us to examine if predictors were differently associated with opioid prescribing by diagnostic group (as shown in results presented on page 10 and in Table 5).  

Comment 12: Presence of an opioid medication prior to hospital admission may be a significant predictor of continued prescription at discharge. Please list the lack of this variable as a limitation. 

Response 12:  The reviewer makes an important point and we now include presence of an opioid medication prior to hospital admission as a limitation (see page 14).

Comment 13: Many limitations are listed, but few strengths of the study are mentioned. Please provide some strengths of the study design here. 

Response 13:  Thank you for this comment. Strengths have been added. These include large sample size, use of a broad sample akin to the typical case mix of PICU admissions, and extraction of data from the EMR (see page 13).

Comment 14: In this section, a clinical implication is offered to increased provider and patient education to pursue alternative treatment strategies for pain, yet the project did not include multi-modal pharmacological pain management strategies as a possible predictor for discharge prescription with opioid. Please either include this variable as a predictor to support the statement (i.e. create flags for whether other non-opioid pain medications were prescribed at discharge and add into your analytical models) or report this construct in the introduction and discussion sections as an important consideration that may reduce need for opioids at discharge. The lack of inclusion of this variable could then be mentioned in the limitation section.

Response 14:  The reviewer brings up an important point – no we do not have data on other multi-modal pharmacological pain management strategies that participants could have been using during their PICU admission may have impacted opioid use or prescription at discharge. We have added this as a limitation to the discussion section (see page 13).

Reviewer 2 Report

Reviewer comments

 This is an interesting study on an interesting topic. It is well writing study on [Who receives an opioid prescription at PICU discharge? Examining clinical and demographic factors associated with receiving an opioid prescription following admission to the pediatric intensive care unit with accept for publication.

1-      The title of the manuscript needs to be summarized, it is too long.

2-      The methods were sufficiently documented.

3-      The statistical methods valid and correctly applied.

4-      The quality of the figures and tables was satisfactory.

5-      The reference list covers the relevant literature adequately.

6-      It is a highly interest to a general audience.

Author Response

Reviewer #2

Comment 1: The title of the manuscript needs to be summarized, it is too long.

                Response 1:  The title has been shortened as requested (see Title page).

Comment 2: The methods were sufficiently documented.

Response 2:  Thank you.

Comment 3: The statistical methods valid and correctly applied.

Response 3:  Thank you.

Comment 4: The quality of the figures and tables was satisfactory.

Response 4:  Thank you.

Comment 5: The reference list covers the relevant literature adequately.

Response 5:  Thank you.

Comment 6: It is a highly interest to a general audience.

Response 6:  Thank you.

Reviewer 3 Report

In this retrospective study of 3345 patients admitted to the PICU, the authors sought to identify both the prevalence of opioid prescriptions at PICU discharge, and the demographic and clinical factors associated with receiving an opioid prescription. They found that 23.7% of all patients discharged to home or rehab were prescribed an opioid. They found that children primarily for surgical reasons were more likely to receive opioid prescriptions, and that certain diagnostic categories (cardiac, neurologic, or hematologic/oncologic) were more likely to have opioids prescribed. They found no differences based on race, ethnicity, sex, or discharge location. Impaired functional status, higher organ dysfunction, and greater number of days on mechanical ventilation were also significantly associated with opioid prescriptions.

The authors are to be congratulated for this interesting and important study. The paper is overall well-written and will require some minor copyediting. Major strengths include the timeliness of the topic and the robust discussion with many recommendations for interventions and future study.

Weaknesses of the paper include conflicting data regarding the importance of patient age, and issues with overall framing. As stated in the results section (but not addressed in the discussion), the age of patient was not statistically significant when examined with chi-square analysis, but emerged as a significant predictor when logistic regression was used. What might explain this discrepancy? Which finding (significant or not) is true? In addition, the introduction frames the importance of this study around PICS-p and PICS-f, based on a potential link between the mental health effects of PICS-p/f and increased risks of opioid prescriptions. However, the study primarily found a link between postoperative surgical admissions and opioid prescriptions. One might suspect different or lower risk for PICS-p/f from planned surgical admissions compared to unexpected medical admissions, which undercuts the framing of importance of this study.

Specific comments:

1.      The word “youth” is used frequently as both singular and plural, which sometimes confuses the syntax of sentences, especially in the results section. Consider using “child” and “children” or pluralizing “youth” as “youths” for ease of understanding.

2.      P1/L37: vaguely worded. Chronic pain increases long-term morbidities after PICU admission? In general? In PICS-p?

3.      P2/L77: “parent mental health may be an additional area of risk” – confusing wording.

4.      Demographics section of Materials: does not explain how ethnicity was categorized or considered, only addresses racial grouping. In addition, many Hispanic families will select “Hispanic” as their ethnicity, and then a different race (as Hispanic is not typically a racial category). Appropriately, the discussion noted that the inclusion of “Hispanic” as a race in the REALD scheme was a limitation in clarity of groups.

a.      Did you consider grouping the race/ethnicity groups as “non-Hispanic white,” “Hispanic,” and “other races”? This might be a more complete description of racial/ethnic groups.

b.      In addition, it is preferable to not center whiteness as the “default” race (as is implied by the grouping into “white/Caucasian” and “non-white/Caucasian”). Best would be to name the specific racial groups you are talking about, but as there are many, “other races” is also acceptable.

5.      P3/L122: These “notes related to participant diagnoses” more likely belong in the discussion section on limitations.

6.       P3/L140: Any explanation for why PELOD scores are grouped 0-1, 2-9, and >10?

7.      P4/L165: needs a new paragraph and another header so this is not thought to be related to functional status at PICU admission

8.      For your tables, having smaller margins and spacing so there is overall less blank space will make it easier to read and follow associated rows.

9.      Table 2 is intended to make the point that prescription patterns vary by age within different diagnostic categories. This is difficult to draw from the table as presented given the different n for each age and subcategory. Adding percentages might make the point more clear?

10.  P7/L221: This sentence contradicts your later discussion regarding importance of age.

11.  Table 3 lists characteristics of opioid and non-opioid prescribed patients. Are there differences between the medical and surgical groups regarding these factors? For instance, did surgical patients have shorter LOS than medical?

12.  In Table 4, consider moving your p value column to the far right so that the confidence interval is presented just after the OR.

13.  P9/L282: missing word “that” before “many”

14.  P10/L306: confusing wording

15.  This paragraph on P10 seems to imply that the outpatient specialty team has not been involved in the patient’s opioid management during hospitalization and therefore needs coordination with the PICU team. This is likely true if the opioids will be managed by a PCP, but if a surgical subspecialty is managing the outpatient care, in many units that team may co-manage during hospitalization as well. The point would be strengthened by acknowledging this and offering a recommendation in that instance.

16.  P10/:329: add “total” before hospitalizations, add hyphen to “physician-assigned”

Author Response

Reviewer #3

General Comments

Comment 1: Weaknesses of the paper include conflicting data regarding the importance of patient age, and issues with overall framing. As stated in the results section (but not addressed in the discussion), the age of patient was not statistically significant when examined with chi-square analysis, but emerged as a significant predictor when logistic regression was used. What might explain this discrepancy? Which finding (significant or not) is true?

Response 1: Thank you for bringing this to our attention so we can add clarity. This discrepancy occurred because a t-test assumes a linear relationship and thus the analysis examined whether the average ages differed by opioid prescribing (yes/no). Given the non-linear relationship between age and opioid prescribing we removed age from the t-test (see page 9). Given logistic regressions are better suited to accommodate non-linear responses, age was retained as a predictor in the regression analyses (see pages 10-11 and Tables 4-5).  We also keep descriptive analyses (presented on page 8 and in Table 2) which show percentages of youth who received an opioid prescription at discharge within each primary diagnostic group, by categorical age groups (0-1, 2-4, 5-10, 11-15).

Comment 2: In addition, the introduction frames the importance of this study around PICS-p and PICS-f, based on a potential link between the mental health effects of PICS-p/f and increased risks of opioid prescriptions. However, the study primarily found a link between postoperative surgical admissions and opioid prescriptions. One might suspect different or lower risk for PICS-p/f from planned surgical admissions compared to unexpected medical admissions, which undercuts the framing of importance of this study.

Response 2: We appreciate the reviewer’s feedback that framing the study within the importance of PICS was potentially confusing and detracted from the focus which was to contribute to the limited research examining predictors of opioid prescribing in youth post PICU-discharge. We have revised the introduction for clarity (see page 3).

Specific comments:

Comment 1: The word “youth” is used frequently as both singular and plural, which sometimes confuses the syntax of sentences, especially in the results section. Consider using “child” and “children” or pluralizing “youth” as “youths” for ease of understanding.

                Response 1: Youth has been replaced with child or children as suggested.

Comment 2: P1/L37: vaguely worded. Chronic pain increases long-term morbidities after PICU admission? In general? In PICS-p?

                Response 2: This sentence has been deleted (in response to “general” comment 2) so no further edits needed.

Comment 3: P2/L77: “parent mental health may be an additional area of risk” – confusing wording.

                Response 3: This sentence has been deleted and the paragraph edited for clarity. 

Comment 4: Demographics section of Materials: does not explain how ethnicity was categorized or considered, only addresses racial grouping. In addition, many Hispanic families will select “Hispanic” as their ethnicity, and then a different race (as Hispanic is not typically a racial category). Appropriately, the discussion noted that the inclusion of “Hispanic” as a race in the REALD scheme was a limitation in clarity of groups.

                Response 4: The REALD asks patients families to identify child race and ethnicity. A participant was considered Hispanic if Hispanic ethnicity was selected. Participants selecting other ethnic identities were categorized as non-Hispanic. Participants who did not select any of the ethnic identity categories were categorized as “unknown”. We recognize that Hispanic individuals may not select Hispanic as a race, for this reason we report both race and ethnicity data. See edits on page 6.

Comment 5: Did you consider grouping the race/ethnicity groups as “non-Hispanic white,” “Hispanic,” and “other races”? This might be a more complete description of racial/ethnic groups.

Response 5: We agree with the reviewer and recognize there are many ways to categorize the data. As authors we had a great deal of discussion on the most accurate way of categorizing/labeling race and ethnicity groups given the limitations of REALD. Given the REALD system includes Hispanic as options for both race and ethnicity we decided it was not accurate to separate “Hispanic” from “other races” as an individual could identify as both Hispanic (ethnicity) and another racial identity and we did not think it was appropriate to re-categorize these choices. While we recognize that the groupings used are imperfect however, as we explain in the discussion section, we were limited by REALD data included in the EMR.

Comment 6: In addition, it is preferable to not center whiteness as the “default” race (as is implied by the grouping into “white/Caucasian” and “non-white/Caucasian”). Best would be to name the specific racial groups you are talking about, but as there are many, “other races” is also acceptable.

Response 6: At the recommendation of the reviewer, we changed our terminology and use “other races” instead of “non-white/Caucasian” in the manuscript.

Comment 7: P3/L122: These “notes related to participant diagnoses” more likely belong in the discussion section on limitations.

                Response 7: We have moved this section from the Methods section to the Discussion. See page 15.

Comment 8: P3/L140: Any explanation for why PELOD scores are grouped 0-1, 2-9, and >10?

                Response 8: We have added a citation to support grouping used in the paper (see page 7).

Comment 9:  P4/L165: needs a new paragraph and another header so this is not thought to be related to functional status at PICU admission

                Response 9: A subheading designating this section as the Statistical Analysis section has been added.

Comment 10: For your tables, having smaller margins and spacing so there is overall less blank space will make it easier to read and follow associated rows.

                Response 10: Where possible we changed margins and spacing as recommended to support readability.

Comment 11: Table 2 is intended to make the point that prescription patterns vary by age within different diagnostic categories. This is difficult to draw from the table as presented given the different n for each age and subcategory. Adding percentages might make the point more clear?

Response 11: Thank you for your feedback. We have discussed this at length and at one point had percentages in the table as suggested by the reviewer. For the submitted manuscript we elected not to include these percentages as they added additional numbers to the already busy table which we felt ended up adding confusion. If the reviewer and editor continue to feel adding the percentages is important we will do.

Comment 12:  P7/L221: This sentence contradicts your later discussion regarding importance of age.

Response 12: Thank you for pointing this out. As noted in response to Reviewer 2 we eliminated this t-test given the non-linear association among age and the outcome variable and appropriately examine age as a predictor of opioid prescribing (yes/no) in the logistic regression models only.

Comment 13: Table 3 lists characteristics of opioid and non-opioid prescribed patients. Are there differences between the medical and surgical groups regarding these factors? For instance, did surgical patients have shorter LOS than medical?

Response 13: Thank you for this thoughtful comment. We ran this analysis and yes there are differences between the surgical and medical groups on LOS. A key reason we included the “surgical admission” variable in regression models was to account for potential variability between the two groups. We believe that further examining differences between primary surgical and non-surgical samples is outside the scope of this manuscript focused on predictors of opioids at discharge but that can be examined in future studies.

Comment 14: In Table 4, consider moving your p value column to the far right so that the confidence interval is presented just after the OR.

                Response 14: The p-value has been moved as recommended.

Comment 15: P9/L282: missing word “that” before “many”

                Response 15: The word has been added.

Comment 16:  P10/L306: confusing wording

                Response 16: This section has been revised for clarity.

Comment 17:  This paragraph on P10 seems to imply that the outpatient specialty team has not been involved in the patient’s opioid management during hospitalization and therefore needs coordination with the PICU team. This is likely true if the opioids will be managed by a PCP, but if a surgical subspecialty is managing the outpatient care, in many units that team may co-manage during hospitalization as well. The point would be strengthened by acknowledging this and offering a recommendation in that instance.

                Response 17: At the reviewer’s recommendation we have acknowledged that in cases of surgical admissions, children may be co-managed during admission and we have added a statement suggesting that collaborative plans among team members could be developed (see page 13).

Comment 18:  P10/:329: add “total” before hospitalizations, add hyphen to “physician-assigned”

Response 18: The sentence has been revised for clarity (see page 14).

Round 2

Reviewer 3 Report

Authors have addressed the critiques sufficiently.